# Rational design of *N*-heterocyclic compound classes via regenerative cyclization of diamines

Robin Fertig[1], Felix Leowsky-Künstler[1], Torsten Irrgang [1] & Rhett Kempe [1] ✉

The discovery of reactions is a central topic in chemistry and especially interesting if access to compound classes, which have not yet been synthesized, is permitted. *N*-Heterocyclic compounds are very important due to their numerous applications in life and material science. We introduce here a consecutive three-component reaction, classes of *N*-heterocyclic compounds, and the associated synthesis concept (regenerative cyclisation). Our reaction starts with a diamine, which reacts with an amino alcohol via dehydrogenation, condensation, and cyclisation to form a new pair of amines that undergoes ring closure with an aldehyde, carbonyldiimidazole, or a dehydrogenated amino alcohol. Hydrogen is liberated in the first reaction step and the dehydrogenation catalyst used is based on manganese.

Reaction discovery is a central topic in chemistry[1] and especially interesting if access to classes of compounds, which have not yet been synthesized, can be provided. Unfortunately, concepts permitting a rational design of compound classes are rare. Iterative synthesis, the regeneration of the functional group(s) originally modified (Fig. 1A), is a suitable tool to introduce chemical diversity, which might be beneficial to address function or global challenges[2]. Recently, metal catalysed reactions have been in focus[2] and used for automated C-C bond formation[3] and selective olefin syntheses employing ethylene[4]. The ring closure of two functional groups generating a new pair of the same functional groups seems an option for synthesizing cyclic compounds (Fig. 1B)[2,5–7]. *N*-Heterocyclic compounds are very important fine and bulk chemicals due to their numerous applications in life and material sciences, for instance, as pharmaceuticals, agro chemicals, dyes, and conductive materials[8]. Classes of *N*-heterocyclic compounds might be accessible if the pair of functional groups that will be regenerated during cyclization are amines (Fig. 1C). We introduce here a catalytic consecutive three-component reaction and classes of *N*-heterocyclic compounds. Our reaction starts with a diamine, which reacts with an amino alcohol via dehydrogenation, condensation, and cyclisation to form a new pair of amines that undergoes ring closure with an aldehyde (Fig. 1C), carbonyldiimidazole or an amino alcohol. Hydrogen is liberated in the first reaction step[9,10] and the dehydrogenation catalyst used is based on the Earth-abundant metal manganese[11–14]. Our reaction

proceeds diastereoselectively, has a large scope, and many functional groups can be tolerated, including hydrogenation-sensitive examples, despite the presence of hydrogen and a hydrogenation catalyst[15]. Upscaling is easily accomplished and a catalytic amount of base is required. All *N*-heterocyclic compounds synthesized here have not yet been reported[16].

## Results

### Reaction optimization

We started our investigations with an optimisation of the reaction conditions of the reaction of 1,8-diaminonaphthalene with 2-aminobenzyl alcohol to form the 2-(2,3-dihydro-1*H*-perimidin-2yl) aniline **A1** (Fig. 2). The synthesis of 2,3-dihydro-1*H*-perimidines from 1,8-diaminonaphtalene and aldehydes is a classic reaction and has been reported already in 1964[17]. Recently, the catalytic generation of the aldehyde for such a coupling via dehydrogenation catalysis employing a phosphine free manganese complex has been reported[18]. The key to our synthesis is the use of amino alcohols to regenerate the set of two amines and we started our investigation with 2-aminobenzyl alcohol. In case of amino alcohols, the corresponding aldehyde can undergo self-condensation and the catalytic generation via dehydrogenation catalysis seems an elegant way to address this issue. Different Earth-abundant metal (Mn, Fe, Co) complexes stabilized by pincer ligands were tested as precatalysts for the dehydrogenation step. Manganese

[1]Lehrstuhl Anorganische Chemie II—Katalysatordesign, Sustainable Chemistry Centre, Universität Bayreuth, 95440 Bayreuth, Germany.
✉e-mail: kempe@uni-bayreuth.de

**A.  Iterative Synthesis**

**B.  Regenerative Cyclization**

Modification degree:          1          2

Termination

**C.  This Work: Synthesis of *N*-Heterocyclic Compounds**

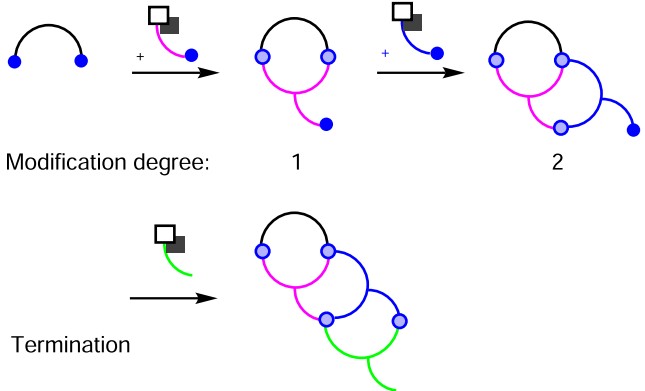

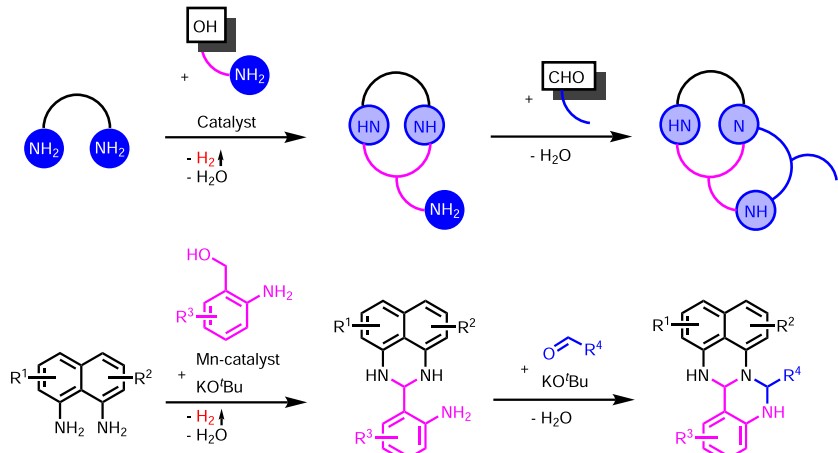

**Fig. 1 | Relevant concepts and work introduced here. A** Regenerating the functional group again that has been modified originally (iterative synthesis) can lead to chemical diversity if different building blocks are used **B** Classes of (poly)cyclic compounds can be conceived via ring closure chemistry. The set of functional groups originally used has to be formed again during the ring closure reaction (regenerative cyclization). Repeating ring closure steps should lead to classes of (poly)cyclic compounds, which have not yet been synthesized, at some stage or modification degree. **C** *N*-Heterocyclic compounds introduced here with amines being the key functional groups, applying a modification degree of two, and a catalytic amino alcohol dehydrogenation-based ring closure reaction as the first step.

catalysts stabilized by a PN₅P-pincer ligand (Fig. 2. top right) showed the highest activity, determined by the yield of the product obtained under the given conditions. Such ligands are easy to synthesize from 2,6-diaminotriazines and dialkyl- or diarylphosphine chlorides. A significantly lower activity was observed if the ligand backbone of the manganese precatalysts was changed from a triazine (PN₅P) to a pyridine (PN₃P) moiety, (precatalysts Mn-**VI**, Mn-**VII**, Supplementary Table 1)[19–22]. Other reaction parameters, such as temperature, precatalyst loading, type and amount of solvent, and base were optimised—see Supplementary Tables 1–7 for details. The optimal reaction parameters for the synthesis of **A1** (Fig. 2) were 1 mol% precatalyst [Mn-**I**], 30 mol% KO*t*Bu, 3 mL 2-MeTHF at 100 °C with a reaction time of 2 h. The reaction proceeded

in a tube with a bubble counter to facilitate the release of hydrogen during the dehydrogenation of the amino alcohol.

## Substrate scope
Regarding the exploration of the functional group tolerance, we used 21 aminobenzyl alcohol derivatives and isolated the corresponding 2,3-dihydro-1*H*-perimidines **A1-A21**, referred to here as amino perimidines for simplification (Fig. 2). The model reaction led to the product **A1** in an isolated yield of 90%. Single crystals were obtained via recrystallization from ethyl acetate/pentane (2:1) at −18 °C and analysed by X-ray diffraction confirming the molecular structure of **A1** (Fig. 2; for more details, see Supplementary Data 1).

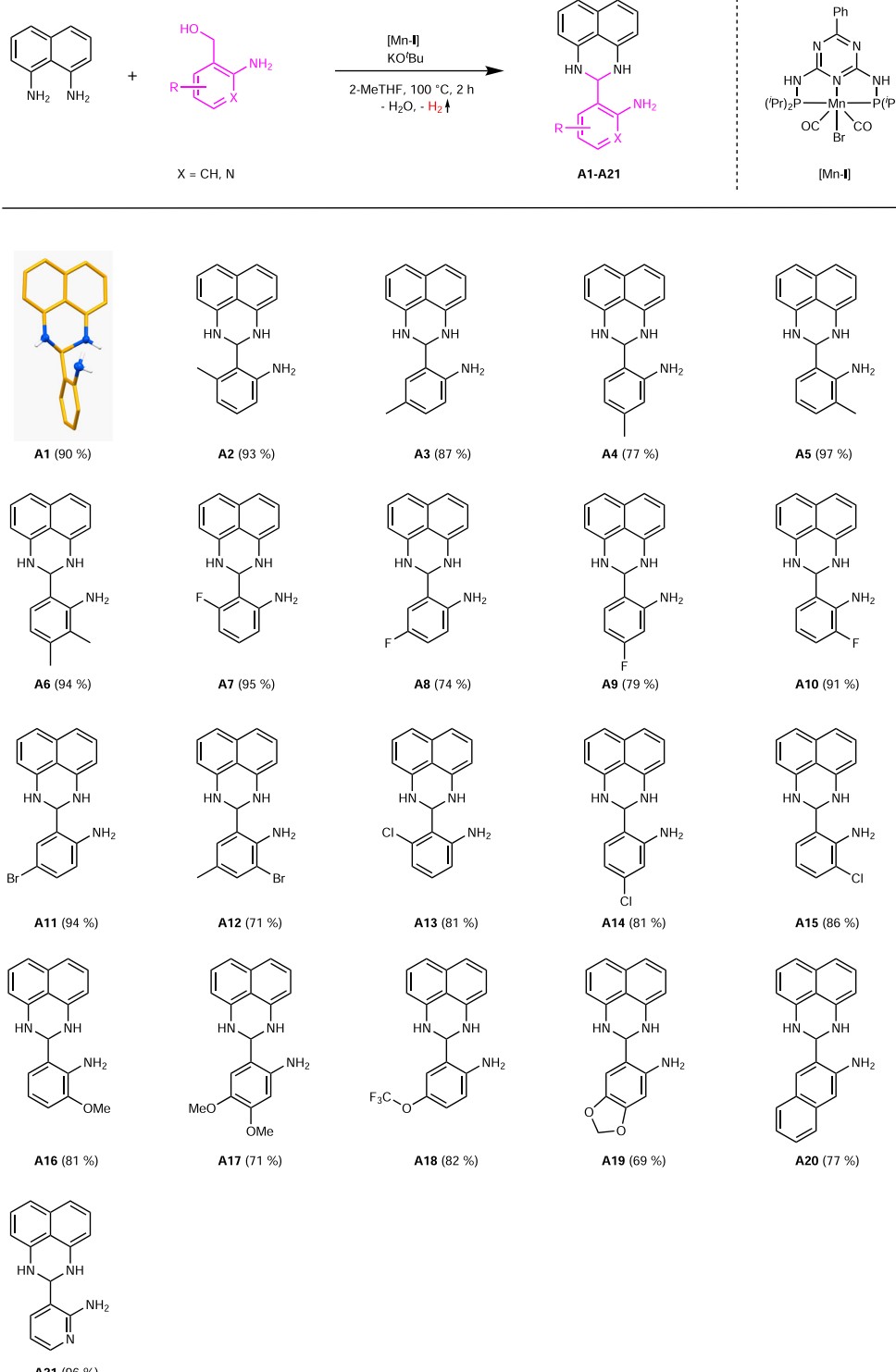

**Fig. 2 | Synthesis of 2,3-dihydro-1*H*-perimidines A1-A21 via liberation of H₂.** Reaction conditions: 2 mmol 1,8-diaminonaphthalene, 2 mmol amino alcohol, 0.6 mmol KO*t*Bu, 1 mol% [Mn-**I**] (0.02 mmol), 3 mL 2-MeTHF, 100 °C (oil bath), 2 h, open system (anaerobic conditions). Isolated yields in brackets.

The products **A2**-**A6** were obtained in yields of 77–97%, demonstrating the tolerance of electron-donating groups on every position at the phenyl substituent. The tolerance of electron withdrawing substituents was shown by using fluoro- (**A7**-**A10**), chloro- (**A13**-**A15**), and bromo-aminobenzyl (**A11**, **A12**) alcohols. The corresponding products were isolated in yields ranging from 71–95%. The fluoro substituent was used as an example to show the tolerance at each position of the phenyl substituent. Substrates containing

methoxy (**A16**), dimethoxy (**A17**), or trifluormethoxy (**A18**) groups were converted smoothly to the products desired and could be isolated in yields up to 82%. An amino perimidine bearing an acetal (**A19**) could be isolated in a yield of 69%. Using a polycyclic aromatic amino alcohol provided **A20** in a yield of 77%. The use of a *N*-heterocyclic amino alcohol led to **A21** in a nearly quantitative yield. The amino perimidines (**A1**-**A24**) were isolated as solids in colours from white to yellow. Each product was not described at that stage. The

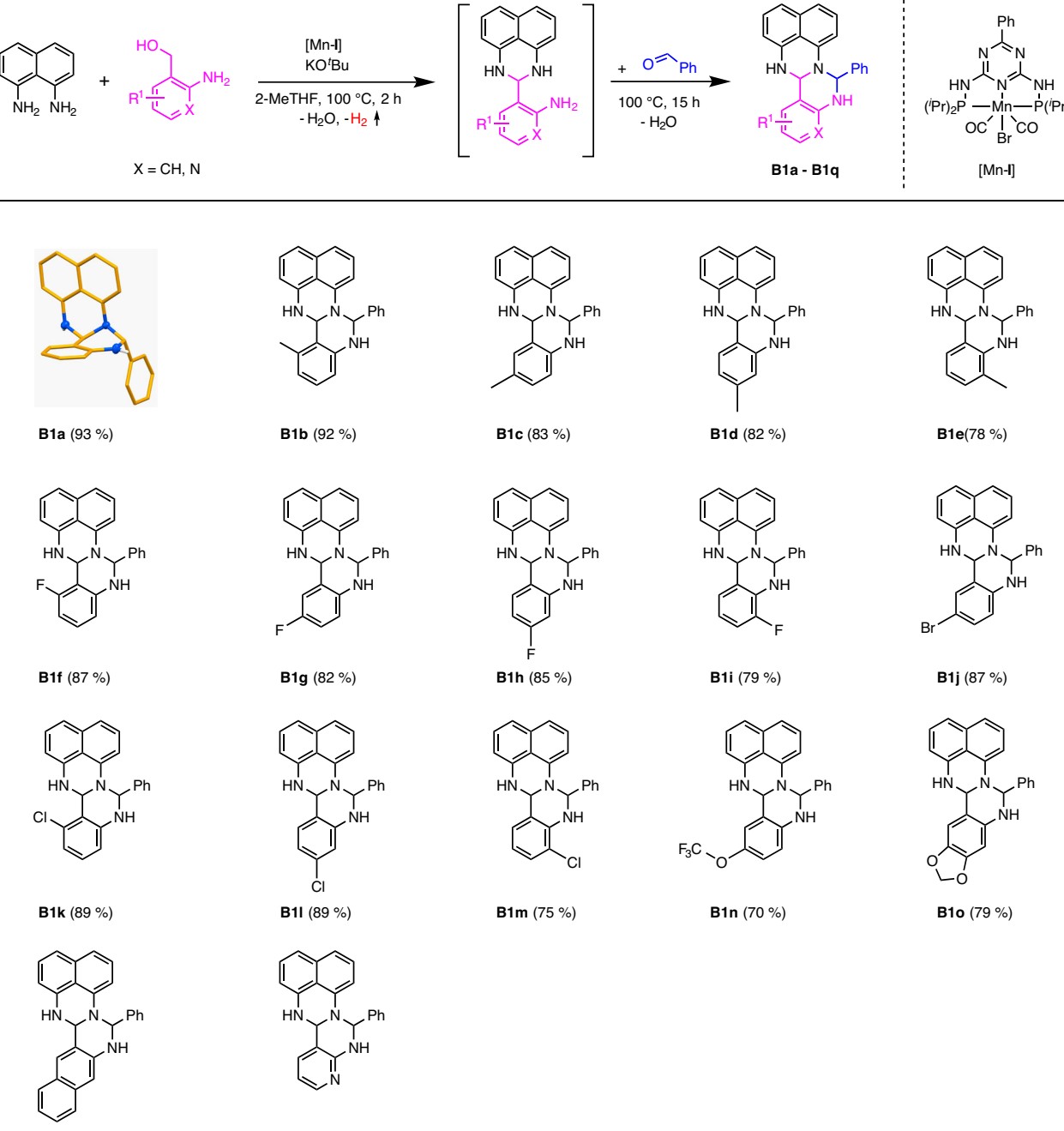

**Fig. 3 | Synthesis of fertigines B1a-B1q: 2-aminobenzyl alcohol variations.**
Reaction conditions: 2 mmol 1,8-diaminonaphthalene, 2 mmol amino alcohol, 0.6 mmol KO𝑡Bu, 1 mol% [Mn-**I**] (0.02 mmol), 3 mL 2-MeTHF, 100 °C (oil bath), 2 h + 15 h, open system (anaerobic conditions). After 2 h: addition of 2 mmol benzaldehyde. Isolated yields in brackets.

amino perimidines generally showed a good solubility in polar solvents, were air-stable, and easy to crystallize (e.g., in ethyl acetate/pentane).

The primary amine functionality of the modification degree 1 and its spatial distance to the NH-groups can be used for a second ring closure (modification degree 2). Aldehydes represent simple, easy-to-handle, inexpensive, diversely available and green or sustainable[23,24] building blocks and can undergo condensation reactions with amines. This modification degree 2 leads to a class of compounds consisting of two six-membered *N*-heterocyclic ring systems (Fig. 3). We propose the name fertigines for this class of *N*-heterocycles. Keeping the synthesis procedure of the fertigines as

simple as possible, we synthesized them via a consecutive multi-component one-pot reaction using the conditions optimised for the synthesis of the amino perimidines followed by the addition of aldehyde (Fig. 3). The addition of benzaldehyde led to the fertigine **B1a** in an isolated yield of 93% after a reaction time of 15 h. **B1a** is a white solid that is soluble in polar solvents. Crystals for single crystal X-ray analysis were obtained by recrystallization of **B1a** (Fig. 3) in ethyl acetate/pentane at −18 °C. The molecular structure of **B1a** is shown in Fig. 3 (for more details, see Supplementary Data 2). The second ring closure proceeded smoothly to the products **B1b-B1e** in yields of 78–92%, indicating no significant influence of the position of electron-donating groups attached to the

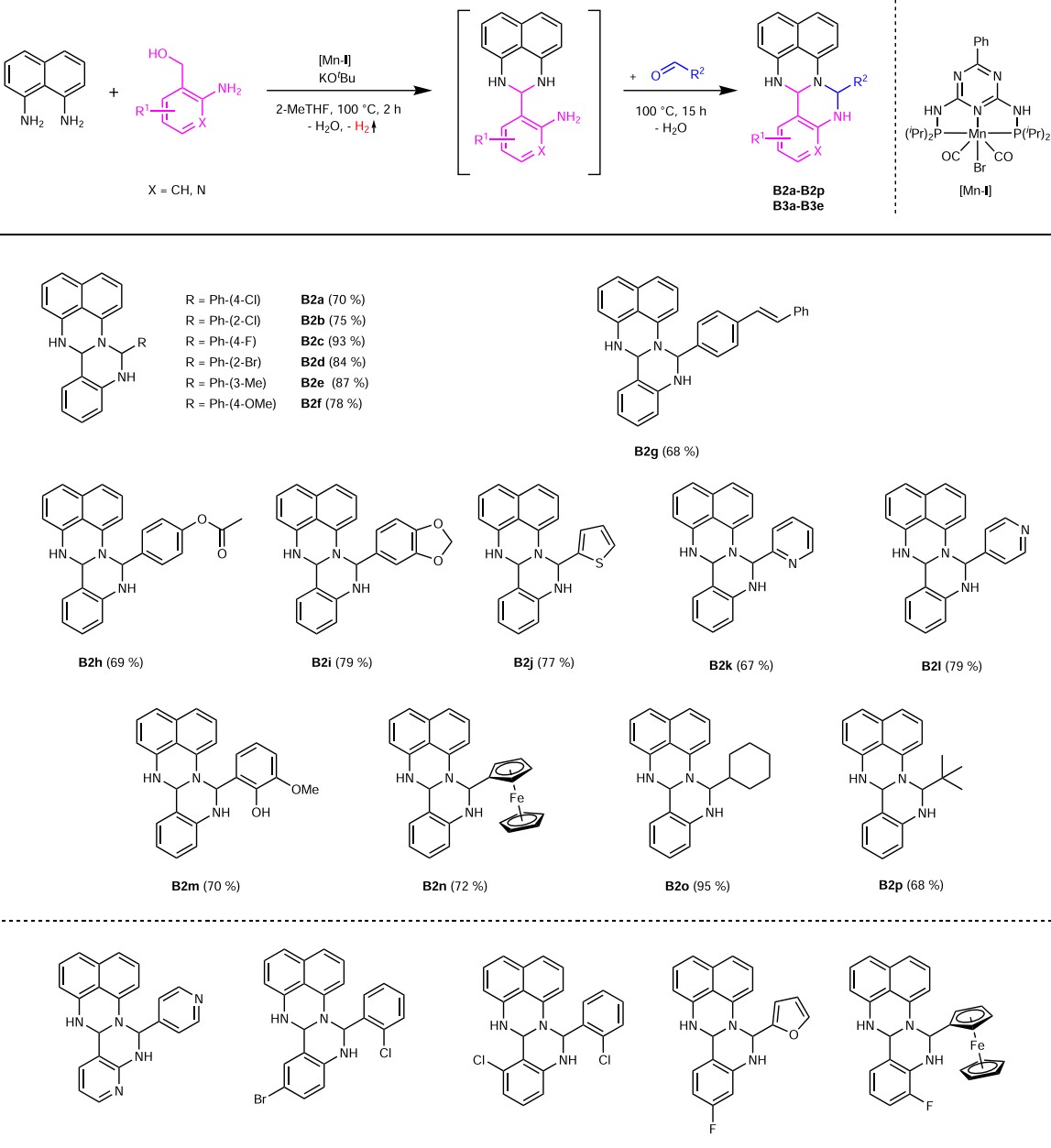

**Fig. 4 | Synthesis of fertigines B2a-B2p and B3a-B3e: aldehyde variations.**
Reaction conditions: 2 mmol 1,8-diaminonaphthalene, 2 mmol 2-aminobenzyl alcohol derivatives, 0.6 mmol KO$^t$Bu, 1 mol% [Mn-**I**] (0.02 mmol), 3 mL 2-MeTHF, 100 °C, 2 h + 15 h, open system (anaerobic conditions). After 2 h: addition of 2 mmol aldehyde. Isolated yields in brackets.

aminobenzyl alcohol moiety. Analogously, we investigated the position-dependent influence of electron-withdrawing groups on the outcome of the reaction. We chose the fluoro substituent and could not observe any significant impact on the second ring closure, obtaining the corresponding fertigines **B1f**-**B1i** in isolated yields of up to 87%. The use of further halogenated substrates, such as 5-bromo- (**B1j**), 6-chloro- (**B1k**), 4-chloro- (**B1l**) or 3-chloro-2-aminobenzyl alcohol (**B1m**), for fertigine synthesis led to the products desired in yields between 75 and 89%. **B1n**, bearing a tri-fluoromethoxy-group, could be obtained in an isolated yield of 70%. A fertigine with an acetal group (**B1o**) on the former amino alcohol moiety was isolated in a yield of 79%. Applying an amino alcohol with a polycyclic aromatic backbone provided the product **B1p** in an isolated yield of 73%. The use of 2-amino-pyridylmethanol resulted in the corresponding product **B1q** in an isolated yield of 86%.

We next investigated the substrate scope of fertigines by using various aldehydes (Fig. 4). After adding benzaldehydes with chloro-substituents in the *para*- and *ortho*-position, we obtained the corresponding fertigines (**B2a**−**B2b**) in isolated yields of 70 - 75%. Other halogenated benzaldehydes, such as *para*-fluorobenzaldehyde or *ortho*-bromobenzaldehyde, reacted smoothly to the corresponding products (**B2c** and **B2d**) and could be isolated in yields of 93 and 84%, respectively. The addition of 3-methylbenzaldehyde to the model reaction (Fig. 2) led to the product **B2e** in a yield of 87%. Methoxy-substituted benzaldehyde provided the corresponding fertigine **B2f**, respectively, in isolated yield of 78%. According to these results, no coherence between the electronic properties of the substituents on benzaldehyde and the efficiency of the second ring closure was observed. Using benzaldehydes for the synthesis of fertigine with a C−C double bond (**B2g**) or an acetoxy group (**B2h**) in the para-

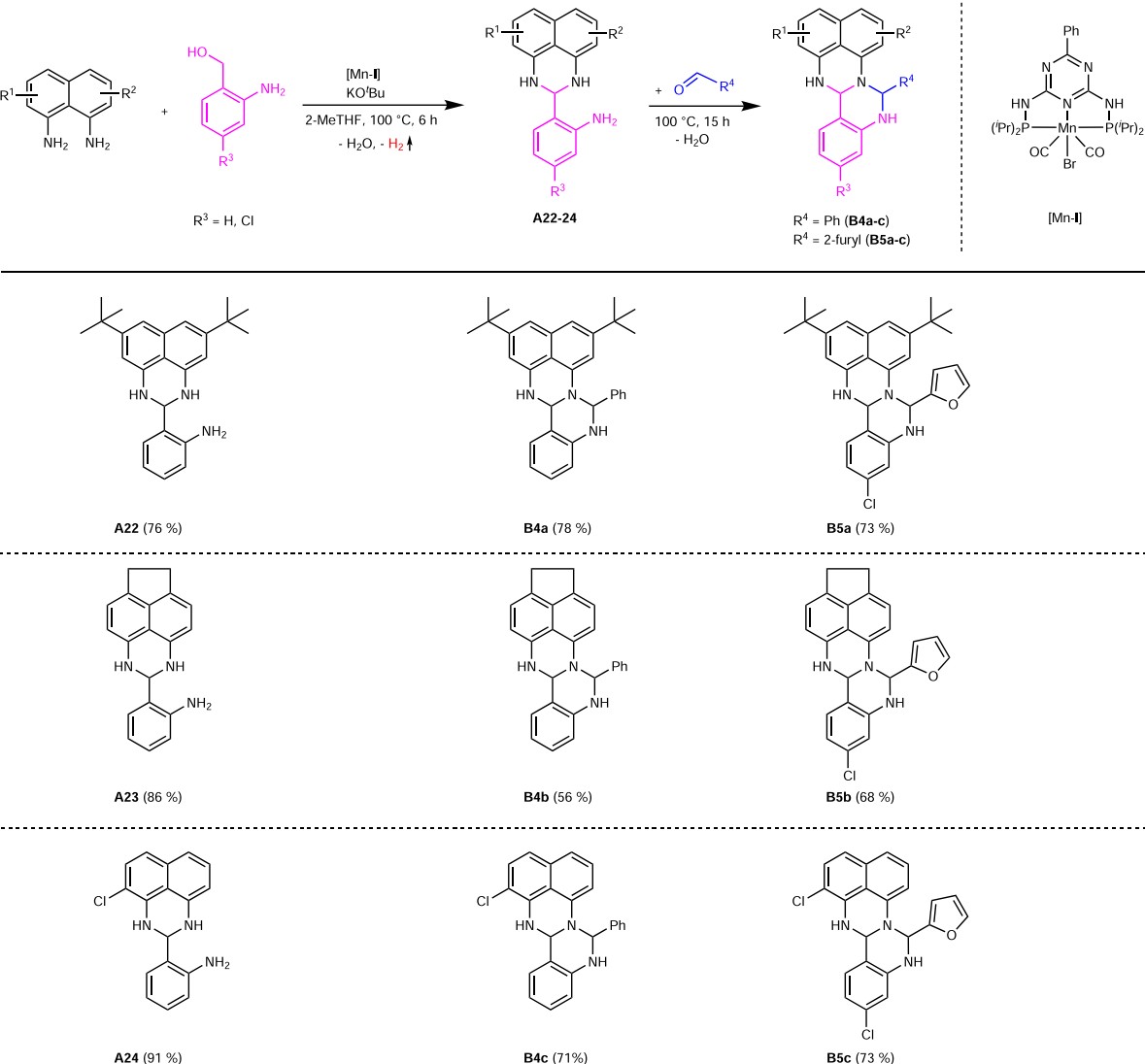

**Fig. 5 | Variation of the diamine for the synthesis of amino perimidines and fertigines.** Reaction conditions: 1 mmol 1,8-diaminonaphthalene derivative, 1 mmol 2-aminobenzyl alcohol, 0.3 mmol KO$^t$Bu, 1 mol% [Mn-**I**] (0.01 mmol), 3 mL 2-MeTHF, 100 °C, 6 h, open system (anaerobic conditions). In order to synthesize the fertigines, 1 mmol aldehyde is added after 6 h. Isolated yields in brackets.

position, the yields decreased to 68 and 69%, respectively, but no notable side reactions occurred. We next investigated several aldehydes with heterocyclic moieties for the synthesis of the corresponding fertigines such as piperonal (**B2i**), thiophen-2-carbaldehyde (**B2j**), 2-formylpyridine (**B2k**) and 4-formylpyridine (**B2l**) and obtained those products in isolated yields up to 79%. The use of *ortho*-vanillin provided **B2m** in an isolated yield of 70%. We also tested an aldehyde based on a metal organic compound, namely, ferrocenaldehyde, and could isolate the fertigine **B2n** in a yield of 72%. The addition of aliphatic aldehydes to the reaction led to fertigines **B2o** and **B2p** in yields of 95 and 68%, respectively. The solubility properties of the fertigines changed using these aldehydes and a good solubility in pentane was observed. There was almost no limitation on the type of aldehyde that could be used for the second ring closure, indicating a very broad scope of our consecutive 3-component reaction. Using an aldehyde and an amino alcohol with a pyridine-backbone, we obtained the fertigine **B3a** in a yield of 90%. Double halogenated fertigines, such as **B3b** or **B3c**, could be isolated in yields of up to 78% by using the corresponding educts. The synthesis of fluorinated fertigines with an *O*-heterocycle (**B3d**) or a metal organic compound (**B3e**) proceeded in yields of 85 and 75%, respectively.

We next addressed the flexibility of the naphthalene diamine in order to achieve a high degree of functionalisation in the resulting fertigines (Fig. 5). Firstly, we investigated the influence of substituted 1,8-diaminonaphthalenes and isolated the resulting amino perimidines **A22**-**A24** (Fig. 5). The use of 3,6-di-*tert*-butyl-1,8-diaminonaphthalene led to the corresponding product **A22** in an isolated yield of 76%. Applying 5,6-diaminoacenaphthene for the catalytic step, an ethylene-bridged naphthalene moiety was achieved and the amino perimidine **A23** was isolated in a yield of 86%. Using 2-chloro-1,8-naphthalenediamine, no decrease in the catalytic activity was observed and the product **A24** was obtained in a yield of 91%. The second modification degree using this 1,8-diaminonaphthalene derivative (**B4a**-**B4c**; **B5a**-**B5c**) was achieved by adding the respective aldehyde after 6 h reaction time. The addition of benzaldehyde led to the products **B4a**-**B4c** desired in yields of up to 78%, observing no significant impact of the naphthalene substitution on the second ring closure. The yield of **B4b** decreased to 56% due to solvation problems. The products **B5a**-**B5c** were isolated in yields from 68–73%.

Upscaling experiments of the model reaction revealed similar yields for amino perimidine as well as fertigine synthesis, obtaining the

**Fig. 6 | Synthesis of amino alkyl perimidines A25-A27[a] and imidazo[1,5-a]peri-midin-10-ones (kuenstlerines) C1-C3.** Reaction conditions: 2 mmol 1,8-diami-nonaphthalene, 2.2 mmol amino alcohol, 0.6 mmol KO[t]Bu, 1 mol% [Mn-I] (0.02 mmol), 12 mL 1,4-Dioxane, 100 °C (oil bath), 4 h, open system (anaerobic conditions). Isolated yields in brackets. [b]Reaction conditions: 2 mmol perimi-dine **A25-27**, 2.3 mmol CDI, 0.6 mmol KO[t]Bu, 10 mL 1,4-Dioxane, 130 °C (oil bath), 2 h, pressure tube (anaerobic conditions). Isolated yields in brackets.

products **A1** and **B1a** desired in multigram scale (Supplementary Information Section 4).

We also examined the reaction of the amino perimidines starting from aliphatic amino alcohols to obtain amino alkyl perimidines (Fig. 6). For this, the reactions of L-alaninol or L-phenylalaninol with 1,8-diaminonaphthalene derivatives were carried out under the optimized conditions for the amino perimidines with only changing the solvent from 2-MeTHF to 1,4-dioxane. The resulting amino alkyl perimidines **A25-A27** were obtained in yields of 91–94% as brown viscous oils and showed a good solubility in polar solvents. Com-pared to the amino perimidines **A1-A21**, the amino alkyl perimidines **A25-A27** are not air stable. Afterwards it was not possible to perform a ring closure reaction between the amino alkyl perimidines **A25-A27** and aldehydes. Therefore we used *N,N'*-carbonyldiimidazole (CDI) as coupling agent and C1 building block to achieve a five-membered *N*-heterocyclic ring. By using a base for the reaction of **A25-A27** with CDI we obtained the corresponding kuenstlerines **C1-C3** (Fig. 6). The optimized reaction parameters for the synthesis of **C1-C3** are 30 mol % KO[t]Bu, 1,4-dioxane as solvent, 1.15 eq. CDI at 130 °C with a reaction time of 2 h in a pressure tube (Supplementary Tables 8–13). We obtained the products desired in yields between 76–91% as light brown to reddish brown solids, which are air sensitive. After the second ring closure, diastereomers were obtained, which can be separated by column chromatography. The diastereomeric ratios varied between 71:29 (**C1**), 88:12 (**C2**), and 61:19 (**C3**). The amino alkyl perimidines **A25-A27** and kuenstlerienes **C1-C3** synthesized here have not yet been reported.

We were also interested in the possibility of synthesizing amino fertigines, from which degree of modification 3 could be achieved. Therefore, we carried out the reactions without further optimization as consecutive one-pot reactions such as for the synthesis of the

fertigines **B1-B5**, and used 2-aminobenzyl alcohols instead of alde-hydes for the second ring closure step (Fig. 7). The amino fertigines **B6a-B6c** were obtained in yields of 38 – 79% as green solids, are poorly soluble in polar solvents and air-stable.

## Mechanistic studies

The mechanism proposed for the catalytic cycle and the ring closure cascade is shown in Fig. 8. The catalyst [Mn-**Ia**] was obtained by adding KO[t]Bu to the precatalyst complex [Mn-**I**][19,20]. The triazine permits the deprotonation of the ligand backbone by strong metal bases, which has been shown to be beneficial in hydrogenation[19,20] and dehy-drogenation catalysis[21,22]. The manganese-catalysed dehydrogenation of 2-aminobenzyl alcohol proceeds via the liberation of one equivalent of hydrogen, as analysed by GC-analysis. In the absence of naphthalene diamine, self-condensation of the 2-aminobenzaldehyde generated in situ took place (Supplementary Fig. 19). We propose the formation of an imine with a subsequent intramolecular ring closure for the amino perimidine synthesis, as revealed by time-dependent [1]H NMR studies. Interestingly, no reaction was observed in the absence of KO[t]Bu, indicating a base-mediated cyclization (Supplementary Figs. 22–24). As the next step, we proposed the in situ deprotonation of one amino functionality of the aminoperimidine obtained by KO[t]Bu (Fig. 8). A yellow crystalline solid (**A1K**) precipitated if KO[t]Bu was added to the amino perimidine **A1** in THF (Supplementary Figs. 25–27). If water was added to **A1K**, it was transformed back to the amino perimidine **A1** accompanied by the formation of KOH (Supplementary Fig. 28). Time-dependent [1]H NMR studies indicate that **A1K** is an intermediate of the second ring closure step (Fig. 8). **A1K** is able to react to the fertigine with benzaldehyde in the absence of KO[t]Bu (Supplementary Figs. 29 and 30) and **A1** doesn't (under analogous conditions).

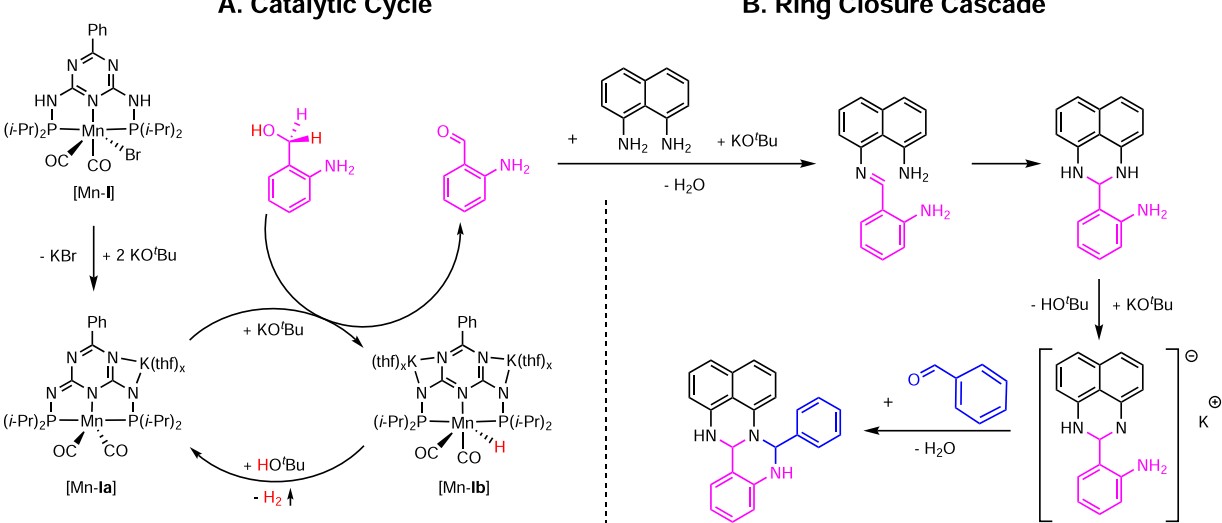

**Fig. 7 | Synthesis of amino fertigines B6a-B6c.** Reaction conditions: 2 mmol 1,8-diaminonaphthalene, 2 mmol 2-aminobenzyl alcohol, 0.6 mmol KO*t*Bu, 1 mol% [Mn-**I**] (0.02 mmol), 4 mL 2-MeTHF, 100 °C (oil bath), 2 h + 15 h, open system (anaerobic conditions). After 2 h: addition of 2.0 or 2.2 mmol 2-aminobenzyl alcohol. Isolated yields in brackets.

**Fig. 8 | Proposed mechanism for the catalytic dehydrogenation and the subsequent ring closure cascade. A** Proposed catalytic cycle. **B** Proposed ring closure cascade.

## Conclusion

The regeneration of a set of diamines via cyclisation of the original set of diamines (regenerative cyclization) permits rational design and the synthesis of novel classes of *N*-heterocyclic compounds. Catalytic amino alcohol dehydrogenation via liberation of hydrogen seems a suitable protocol to accomplish regenerative cyclization of diamines extending the existing amino alcohol dehydrogenation based *N*-heterocycle syntheses, for instance, the synthesis of pyrroles[25,26] and pyridines[27,28]. Recent work of cyclization of diamines employing methanol[29] holds promises for the generalization of the concept introduced here.

## Methods

### General procedure for the synthesis 2-aminophenyl-2,3-dihydro-perimidines (1) and fertigines (2)

In a glovebox, 2 mmol 1,8-naphthalenediamin and 2 mmol 2-aminobenzyl alcohol derivatives are dissolved in 1 mL 2-MeTHF and added to a Schlenk tube. 0.5 mL of a 0.04 mmol/mL stock solution of the Mn-precatalyst Mn-**I** and 0.5 mL of a 1.2 mmol/mL KO*t*Bu stock solution are added to the Schlenk tube. 1 mL 2-MeTHF is added, and the reaction mixture is heated at 100 °C using an open system consisting of a reflux condenser and a bubble counter. **(1):** After 2 h, the reaction is stopped by cooling down to room temperature and the addition of 2 mL H$_2$O. Depending on the product, two different methods for purification were performed: 1. The mixture is extracted with dichloromethane (3 × 10 mL), the organic layers are dried with Na$_2$SO$_4$ and the solvent is removed in vacuo. The crude product is purified by column chromatography using Alox N as stationary phase. 2. H$_2$O (5 mL) is added, the product is precipitated with pentane, filtrated, and washed with pentane. Finally, it is dried in vacuo. **(2):** After 2 h, 2 mmol of various aldehydes (dissolved in 0.5 mL 2-MeTHF) are added to the reaction. After 15 h, the reaction is stopped by cooling down to room temperature. The work-up depends on the substrates used. Usually, 2 mL H$_2$O is added, and the reaction mixture

is extracted with dichloromethane (3 × 10 mL). The organic layers were dried with $Na_2SO_4$ and the solvent is removed in vacuo. The crude product is purified by column chromatography using Alox N as stationary phase.

## General procedure for the synthesis of 7,7a,8,9-tetrahydro-10H-imidazo[1,5-a]-perimidin-10-one derivatives

In a glovebox, 2 mmol perimidine derivatives, 30 mol% KO$^t$Bu (0.6 mmol, dissolved in 1.5 mL 1,4-dioxane), and 2.3 mmol carbonyldiimidazol (CDI) are added to a pressure tube, and dissolved in 8.5 mL 1,4-dioxane. The sealed pressure tube is heated at 130 °C for 2 h. After cooling down to room temperature 30 mL water is added and the product is extracted with diethyl ether (4 × 50 mL). The organic layers are dried with $Na_2SO_4$ and the solvent was removed in vacuo. The crude product is purified via gradient column chromatography using Alox N as stationary phase.

## Data availability

Crystallographic data for compounds **A1** and **B1a** are available free of charge from the Cambridge Crystallographic Data Centre under references CCDC 2084882 and CCDC 2083140, respectively. Materials and methods, experimental procedures, mechanistic studies, characterization data, and spectral data are available in the Supplementary Information. Correspondence and requests for materials should be addressed to R.K.

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

## Acknowledgements

We thank Heidi Maisel and Felix Schreiner for their support in the lab. We acknowledge financial support of the DFG KE 756/31-2.

## Author contributions

R.K. conceived the concept. R.F., F.L.-K., T.I., and R.K. jointly devised the experimental program. T.I. supervised the experimental program. R.F. and F.L.-K. carried out the experimental program. All authors jointly wrote the manuscript.

## Funding

## Competing interests

The authors declare no competing interest.
