## [Peer Review File · Nature Communications]

REVIEWER COMMENTS

Reviewer #1 (Remarks to the Author):

This paper describes a highly interesting system for the synthesis of multi-N-heterocyclic compounds. A large number of new compounds, which have not been reported, were successfully synthesized from easily-available starting materials. Thus, this work is of great impact in the research field of synthetic organic chemistry. Additionally, the concept of "iterative synthesis", which is the basis of this work, is noteworthy. The manganese catalysts employed in this work have been originally developed by the author's group.

The content of this paper seems to have sufficient novelty and urgency suitable for the publication in high-quality journal. This paper is concisely written, and the results are clearly indicated in the main text and supplementary information.

Therefore, this referee is supportive to accept for publication in Nature Communications.

However, the authors should address to the following comments, and revise the manuscript appropriately.

1. For a considerable number of new compounds, the results of elemental analysis are not satisfactory. Authors should consider the results of the analysis. Further purification should be performed for several compounds.
2. Experimental data for the qualitative and quantitative analyses of evolved hydrogen during the reactions should be indicated. A very brief explanation is given on page S16 in the supplementary information, but a figures or charts showing the measurement results should be added.
3. The reaction of A1 with 2-aminobenzyl alcohol should be examined. If this reaction proceeds well, it might be possible to reach modification degree 3. There could be very interesting developments, so why not try this reaction?
4. In the first section of this paper, there is an expression "PN5P-pincer". Considering the wide range of readers of this journal, this referee thinks it would be better to provide a little more explanation.
5. Number of citations seems to be rather small. Please cite literature on catalytic carbon-nitrogen bond formation and alcohol dehydrogenation.

Reviewer #2 (Remarks to the Author):

In this work, Kempe and coworkers demonstrated the synthesis of heterocycles via three-component coupling of 2-aminobenzyl alcohols, 1,8-aminonaphthalenes, and aryl aldehydes. The reaction was catalyzed by an Earth's abundant manganese complex having PN5P-pincer backbone. The scope was broad, tolerating a range of electronically and sterically biased functional groups and heterocycles. However, the major concern is that the first step of this reaction, i.e., the synthesis of perimidine derivatives from the dehydrogenative coupling of 2-amino benzyl alcohols and 1,8-aminonaphthalenes is already explored with an (NNS)Mn(I)-complex (*Organometallics* 2019, 38, 1815-1825). The present work utilized 2-aminobenzyl alcohol as a coupling partner and is a subset of the above-mentioned original publication. Besides, the present manuscript required an alkyl-phosphine-derived catalyst, almost at the same loading, albeit at a slightly lower temperature. The concept and the mechanism, however, are the same. The second step is a trivial condensation of 1,3-diamine with aryl aldehydes, again just further derivatization of perimidines. Therefore, the methodology lacks novelty and is unsuitable for the high standard of Nature communication. Considering the amount of work on heterocycle synthesis, the manuscript can be published in a specialized journal.

Minor comments:

- 1) The authors did not explore the possibility of extending the scopes to aliphatic 1,3-amino alcohols. Could they check if the pathway is viable with aliphatic alcohols too?
- 2) The intensity of the ¹³C spectra for the majority of the compounds is too low to be visible and contains impurity.

Reviewer #3 (Remarks to the Author):

In this manuscript, Kempe and co-workers described a consecutive three-component reaction for synthesizing fertigines using an efficient manganese catalyst based on a PN5P-pincer ligand. This protocol involves the acceptorless dehydrogenation of amino alcohols catalyzed by manganese to form amino aldehydes, and the subsequent synthesis of amino perimidines with diamines, which are useful synthons and can be easily isolated, and finally undergo the ring closure cascade with aldehydes. The substrate scope is satisfactory, many functional groups can be tolerated, including hydrogenation-sensitive examples, such as carbon-carbon double bond and ester groups. Notably,

the authors showed that the reaction was easily scaled up to 15 mmol without erosion of the yield. Overall, I recommend the acceptance of this manuscript to be published on Nature Communications after minor revision.

My comments and questions are as follows:

1. Catalytic dehydrogenation transformation of alcohols has always been a research hotspot in recent years, and new examples are constantly being reported. In this regard, the recent research work (Chin. J. Chem. 2022, 40, 1137.) is relevant to this article, which is recommended to be cited it in this manuscript.
2. In general, a summary is desired to summarize the manuscript and outlook. Therefore, it is suggested to add a summary and outlook at the end of this manuscript.
3. In Supplementary materials, ¹⁹F NMR spectra of products A7-10, A18, B1f-i, B1n, B2d and B3d-e were not attached, and relevant data were missing. Important data of these new compounds need to be collated in Supplementary materials for reference.

Response to Reviewers

“Reviewer: #1 (Remarks to the author)

This paper describes a highly interesting system for the synthesis of multi-N-heterocyclic compounds. A large number of new compounds, which have not been reported, were successfully synthesized from easily-available starting materials. Thus, this work is of great impact in the research field of synthetic organic chemistry. Additionally, the concept of "iterative synthesis", which is the basis of this work, is noteworthy. The manganese catalysts employed in this work have been originally developed by the author's group.

The content of this paper seems to have sufficient novelty and urgency suitable for the publication in high-quality journal. This paper is concisely written, and the results are clearly indicated in the main text and supplementary information.

Therefore, this referee is supportive to accept for publication in Nature Communications.

However, the authors should address to the following comments, and revise the manuscript appropriately.”

Our response: Thank you very much to Reviewer 1 for evaluating and improving our manuscript.

“1. For a considerable number of new compounds, the results of elemental analysis are not satisfactory. Authors should consider the results of the analysis. Further purification should be performed for several compounds.”

Our response: Thank you. We went through all of the EA. A few of them have deviations larger than 0.7 %, the error of our analyzer. If so, we purified again and did EA again.

Our alteration: The corresponding results were listed in the SI.

“2. Experimental data for the qualitative and quantitative analyses of evolved hydrogen during the reactions should be indicated. A very brief explanation is given on page S16 in the supplementary information, but a figures or charts showing the measurement results should be added.”

Our response: Thank you.

Our alteration: We added the data requested to the SI.

“3. The reaction of A1 with 2-aminobenzyl alcohol should be examined. If this reaction proceeds well, it might be possible to reach modification degree 3. There could be very interesting developments, so why not try this reaction?”

Our response: Thank you. Very interesting suggestion. We did it, see Table 6. We reacted the starting diamine with 2 equivalents of 2-aminobenzyl alcohol or in two other examples with 1 equivalent of two different 2-aminobenzyl alcohols sequentially.

Our alteration: We added the corresponding results to the manuscript (Table 6) and the SI.

“4. In the first section of this paper, there is an expression "PN5P-pincer". Considering the wide range of readers of this journal, this referee thinks it would be better to provide a little more explanation.”

Our response: Thank you.

Our alteration: We added an explanation as suggested.

“5. Number of citations seems to be rather small. Please cite literature on catalytic carbon-nitrogen bond formation and alcohol dehydrogenation.”

Our response: Thank you.

Our alteration: Beside the publications discussed below (vide infra, reviewer #2 and #3) and additional citations in the conclusion and outlook section, we now cite the first examples of Mn catalyzed dehydrogenative C-N-bond formation (Milstein and coworkers JACS 2016) and Kirchner and coworkers (Chem. Eur. J 2016) and the first example of the Mn catalyzed alkylation of amines by alcohols Beller and coworkers (Nature Commun. 2016).

“Reviewer: #2 (Remarks to the author)

In this work, Kempe and coworkers demonstrated the synthesis of heterocycles via three-component coupling of 2-aminobenzyl alcohols, 1,8-aminonaphthalenes, and aryl aldehydes. The reaction was catalyzed by an Earth’s abundant manganese complex having PN5P-pincer backbone. The scope was broad, tolerating a range of electronically and sterically biased functional groups and heterocycles.”

Our response: Thank you very much to Reviewer 2 for evaluating and improving our manuscript.

“However, the major concern is that the first step of this reaction, i.e., the synthesis of perimidine derivatives from the dehydrogenative coupling of 2-amino benzyl alcohols and 1,8-aminonaphthalenes is already explored with an (NNS)Mn(I)-complex (*Organometallics* 2019, 38, 1815-1825). The present work utilized 2-aminobenzyl alcohol as a coupling partner and is a subset of the above-mentioned original publication. Besides, the present manuscript required an alkyl-phosphine-derived catalyst, almost at the same loading, albeit at a slightly lower temperature. The concept and the mechanism, however, are the same. The second step is a trivial condensation of 1,3-diamine with aryl aldehydes, again just further derivatization of perimidines. Therefore, the methodology lacks novelty and is unsuitable for the high standard of Nature communication. Considering the amount of work on heterocycle synthesis, the manuscript can be published in a specialized journal.”

Our response: Thank you. We disagree with the statement of reviewer #2 “the synthesis of perimidine derivatives from the dehydrogenative coupling of 2-amino benzyl alcohols and 1,8-aminonaphthalenes is already explored with an (NNS)Mn(I)-complex (*Organometallics* 2019, 38, 1815-1825)”. In *Organometallics* (2019, 38, 1815-1825), a beautiful piece of work indeed, the authors describe an interesting Mn catalyst for the N-alkylation of amines by alcohols, the synthesis of imines from amines and alcohols and the synthesis of 2,3-dihydro-1h-perimidines from alcohols and 1,8-diaminonaphthalene. No coupling of 2-amino benzyl alcohols or any amino alcohol and 1,8-diaminonaphthalene was reported. The key to our synthesis is the use of amino alcohols to regenerate the set of two amines. The synthesis of 2,3-dihydro-1h-perimidines from 1,8-diaminonaphthalene and aldehydes is a classic reaction and has been reported already 1964 by F.D. Popp and A. Catala (*J. Heterocyclic Chem.* 1964, 1, 108-109). Since the coupling of an 1,8-diaminonaphthalene and aldehydes is classic, we used it as the first part of our multi-component sequence.

We also feel that the statement: “Besides, the present manuscript required an alkyl-phosphine-derived catalyst, almost at the same loading, albeit at a slightly lower temperature” is a bit questionable. We use 1 mol% catalyst, 100 °C and 2h, in *Organometallics* (2019, 38, 1815-1825), the authors use 5 mol% catalyst, 140 °C and 24 h. An advantage of the ligand we use is its extremely simple synthesis. However, the concept of phosphine-free ligands is also very appealing.

We are very sorry for not discussing and citing *Organometallics* (2019, 38, 1815-1825) and the original work of Popp and Catala from 1964. Again, I want to congratulate the authors for this beautiful piece of work. For us, the amino alcohol dehydrogenation is the key to make to concept work and so we neither cited the *Organometallics* paper of Srimani and coworkers and Srivastava nor the pioneering and classic work of Popp and Catala. A completely agree with the authors that this has to be changed.

Our alteration: We now cite and discuss Organometallics (2019, 38, 1815-1825) and the work of Popp and Catala from 1964 (J. Heterocyclic Chem. 1964, 1, 108-109).

“Minor comments:

1) The authors did not explore the possibility of extending the scopes to aliphatic 1,3-amino alcohols. Could they check if the pathway is viable with aliphatic alcohols too?”

Our response: Thank you. Very good suggestion. We included aliphatic amino alcohols and another ring closure reaction. Six new examples have been synthesized. See Table 5.

Our alteration: Results were added to the manuscript and experimental details to the SI.

“2) The intensity of the ^{13}C spectra for the majority of the compounds is too low to be visible and contains impurity.”

Our response: Thank you. We have revised the ^{13}C spectra accordingly.

Our alteration: We have added the revised spectra to the SI.

“Reviewer: #3 (Remarks to the author)

In this manuscript, Kempe and co-workers described a consecutive three-component reaction for synthesizing fertigines using an efficient manganese catalyst based on a PN5P-pincer ligand. This protocol involves the acceptorless dehydrogenation of amino alcohols catalyzed by manganese to form amino aldehydes, and the subsequent synthesis of amino perimidines with diamines, which are useful synthons and can be easily isolated, and finally undergo the ring closure cascade with aldehydes. The substrate scope is satisfactory, many functional groups can be tolerated, including hydrogenation-sensitive examples, such as carbon-carbon double bond and ester groups. Notably, the authors showed that the reaction was easily scaled up to 15 mmol without erosion of the yield. Overall, I recommend the acceptance of this manuscript to be published on Nature Communications after minor revision.”

Our response: Thank you very much to Reviewer 3 for evaluating and improving our manuscript.

“My comments and questions are as follows:

1. Catalytic dehydrogenation transformation of alcohols has always been a research hotspot in recent years, and new examples are constantly being reported. In this regard, the recent research work (Chin. J. Chem. 2022, 40, 1137.) is relevant to this article, which is recommended to be cited it in this manuscript.”

Our response: Thank you. Very good suggestion. This is a very inspiring manuscript.

Our alteration: We cite and discuss this publication.

“2. In general, a summary is desired to summarize the manuscript and outlook. Therefore, it is suggested to add a summary and outlook at the end of this manuscript.”

Our response: Thank you.

Our alteration: We added an abstract and a conclusion and outlook section.

“3. In Supplementary materials, ¹⁹F NMR spectra of products A7-10, A18, B1f-i, B1n, B2d and B3d-e were not attached, and relevant data were missing. Important data of these new compounds need to be collated in Supplementary materials for reference.”

Our response: Thank you.

Our alteration: We added as suggested.

REVIEWERS' COMMENTS

Reviewer #1 (Remarks to the Author):

The authors have improved the manuscript appropriately as requested.

I think that the revised version is now acceptable for publication in Nature Communications.

Reviewer #2 (Remarks to the Author):

In the revised manuscript, the authors have responded to all the minor questions raised by the referees. However, the response to referee 2's comments is unsatisfactory. The work is a corollary to the previous report on the dehydrogenative coupling of alcohols with 1,8-aminonaphthalenes. The present manuscript used 2-amino benzylic alcohol as a substrate instead of benzylic alcohol. The resulting product is then condensed with aldehydes, uncatalyzed, under heating. The reviewer believes it does not bring enough novelty to be published in this journal. The manuscript is just a combination of two known reactions. It is more like a demonstration of the synthetic applicability of these known reports instead of a stand-alone publication in this high-impact journal.